# Exploring Consumers’ Purchase Intention of an Innovation of the Agri-Food Industry: A Case of Artificial Meat

**DOI:** 10.3390/foods9060745

**Published:** 2020-06-04

**Authors:** You-Cheng Shen, Han-Shen Chen

**Affiliations:** 1Department of Health Diet and Industry Management, Chung Shan Medical University, Taichung City 40201, Taiwan; youcheng@csmu.edu.tw; 2Department of Nutrition, Chung Shan Medical University Hospital, Taichung City 40201, Taiwan; 3Department of Medical Management, Chung Shan Medical University Hospital, Taichung City 40201, Taiwan

**Keywords:** sustainable food systems, global change, dietary diversity, food environments, consumer behavior

## Abstract

Green consumption is an emerging environmental topic receiving global attention. Because livestock production is a primary source of greenhouse gas emission, the “low-carbon diet” has become a new trend in the catering industry. Fast food companies have been launching vegetarian products because artificial meat requires less water and land resources than traditional livestock and has lower carbon emissions. This study explores the influence of consumers’ attitude, subjective norms (SNs), and perceived behavioral control (PBC) on their purchase intention for vegetarian burgers from the product knowledge (PK) and environmental concern (EC) perspectives. Based on the theory of planned behavior, the purchase intention of people from different food cultures to pay for fast food burgers is discussed. Five hundred questionnaires were distributed, of which 436 were valid. The results revealed that: (1) consumers’ SNs, PBC, and EC significantly affect purchase intention (PI), and SNs and PK have no significant relationship with PI; and (2) vegetarians are willing to pay higher prices than nonvegetarians. This study recommends that industry personnel should attempt to increase consumers’ knowledge regarding artificial meat and expand marketing channels to improve the convenience of purchasing artificial meat foods by conducting lectures and media promotion, respectively.

## 1. Introduction

In recent years, climate change has resulted in a number of serious global threats. Global warming is primarily caused by excessive greenhouse gas (GHG) emissions, including that of CO_2_, CH_4_, N_2_O, etc. According to the Fourth Assessment Report (AR4) of the Intergovernmental Panel on Climate Change (IPCC) [1], several human activities can produce GHG emissions; some of its sources are energy supply, industrial activities, forestry (including deforestation), agriculture, transportation, commercial buildings and residential buildings, garbage, and sewage. Most of the GHGs generated in diet-related areas are produced by fossil fuels and agricultural activities. The GHGs produced in agricultural activities account for 13.5% of all the GHG emissions, which is more than the global GHG emissions from transportation. The Food and Agriculture Organization of the United Nations (FAO) [2] found that the GHGs generated in the livestock industry comprise of 80% of the agricultural emissions and are the chief source of agricultural GHG emissions [2]. However, according to FAO’s survey report [3], the global population will reach 9 billion people by 2050, and the global food demand will increase by 70%; the demand for meat such as poultry, pork, and beef will increase by 73% [3]. The ethical issues and environmental problems related to meat production will be exacerbated [4] because livestock products are considered to be the primary cause of GHG emissions [5]. Marlow et al. noted that in a nonvegetarian diet, there is 2.9 times more water consumption, 2.5 times more energy consumption, 13 times more fertilizer consumption, and 1.4 times more pesticide consumption than that in a vegetarian diet; the differences are chiefly caused by the inclusion of beef in the diet [6]. Therefore, dietary changes will reduce GHG emissions and contribute to the reduction of global warming [7,8]. Cultured or artificial meat is a new solution for clean proteins; it can be produced from animal cells through a cultured medium rather than from slaughtered animals [9,10]. Artificial meat largely circumvents the need for animals in the meat production system, thus alleviating a milieu of animal welfare, public health, and environmental concerns (EC) associated with conventional meat [11,12,13].

With the increase in global awareness regarding the need for carbon reduction and environmental protection, fast food companies have launched vegetarian products because artificial meat requires less water and land resources than traditional livestock and results in lower carbon emissions. Since 2017, McDonald’s has been launching vegan burgers and peripheral vegetarian products successively in places such as Finland, Sweden, and France, as well as several vegetarian products in response to the 40% vegetarian population in India. In January 2019, it introduced a full vegetarian meal for children and a spicy vegetarian burrito meal. In April 2019, Burger King in collaboration with Impossible Foods, an artificial meat company, produced vegetarian meat burgers, replacing beef with plant-based meat fillings, which increased Burger King’s sales by 18.5%. There are few studies based on the behaviors of consumers of artificial meat [14,15,16,17,18]; therefore, the author was motivated to conduct this study.

The theory of planned behavior (TPB) was employed to explain and predict consumer behavior patterns in specific situations. Since its development, this theory has been widely employed by scholars in the research area of consumers’ purchase intention [19,20,21,22,23,24], which reveals its significance. For example, Yazdanpanah and Forouzani used TPB to explore Iranian students’ intention to purchase organic foods with other research variables such as ethics and self-identity combined [23]. Yadav and Pathak used TPB to study green consumption behavior in developing countries [25]. Wong et al. used TPB to explore consumer’s intention to buy suboptimal foods [22]. Chang et al. used TPB to investigate consumers’ willingness to purchase insect foods [20]. The main contribution of the paper is providing the first findings regarding the consumer behavior of a polemic but sustainable protein alternative of meat.

Studies have shown that product knowledge (PK) can affect consumers’ purchasing behavior [26,27,28]. Consumers’ level of knowledge and whether they understood the information easily influences their purchasing decisions [26]. Previous studies mostly used measures such as subjective, objective, and experience-based knowledge [28,29]. In this study, because artificial meat products are still relatively novel to Taiwanese consumers, each consumer had different PK levels, which may have lead them to possess different intentions to buy artificial meat products. Therefore, this study included PK as one of its variables.

With the change in consumption behavior caused by an increase in EC, the individual’s responsibility for environmental protection and the consumption behavior pattern for commodities have received increasing attention [24]. Dunlap and Van Liere first proposed the New Environmental Paradigm (NEP) Scale [30], and then Dunlap made further amendments in order to assess people’s environmental values and ethics [31]. Studies have found that consumers do not simply make purchases on the basis of their preferences; instead, they have begun valuing low-carbon diets; local food ingredients; and seasonal, organic, and fair-trade products; as well as paying attention to product labels [32,33]. Consumers’ EC significantly impacts the purchase of green products [21,34]. Therefore, this study included EC as a study variable.

In addition, consumers’ product choices involve elements such as their perceptions, expectations, social and psychological factors, financial circumstance, and intrinsic/external product characteristics [35]. Among them, price is a decisive factor that influences purchase behavior among external characteristics [36]. Ling’s study found that green products are more expensive than ordinary products because of the higher cost of the production process [37]. Rex and Baumann argued that consumers are willing to pay higher prices for green products [38]. When consumers are positively concerned about the environment, they are willing to pay a higher price to buy goods causing lower levels of environmental damage to engage in environment-friendly behaviors [39,40]. When discussing consumers’ willingness to pay (WTP) for green products, the contingent valuation method (CVM) is one of the most commonly used assessment methods. Kang et al.’s study showed that 37% of consumers were willing to pay an additional fee of 1%–5%, 24% of consumers were willing to pay an additional fee of 6%–10%, and 5.5% of consumers were willing to pay an additional fee of 10% for consumption in green restaurants [41]. Liu et al. explored consumers’ WTP for sustainable coffee products in accordance with demographic variables and concluded that the WTP of respondents who were over 65 years old, had a master’s degree or higher, and earned more than 90,000 yuan per month were significantly higher than that of the other respondents [42].

In summary, on the basis of TPB, this study examines the influence of ATT, SNs, and PBC on the PI of customers. Then, PK and EC are added to study Taiwanese customers’ perceptions and viewpoints on artificial meat foods. In addition, the WTP of different dietary culture groups for artificial meat foods is discussed. Finally, the results of the study will provide the food or catering industry operators with a basis for formulating business management strategies.

The rest of the paper is arranged as follows: Section 2 covers the literature on TPB, PK, and EC, presents the hypotheses, and highlights the relationships among the hypotheses. Section 3 explains the research methodology, which encompasses data collection, model construction, and measurement. Then, Section 4 presents the data analysis process, including structural equation modeling (SEM) and single-bound dichotomous choice model. Finally, Section 5 discusses the research limitations and suggestions for future research are expressed.

## 2. Review of Literature and Hypothesis Development

### 2.1. Theoretical Framework

The study adopted TPB and added two constructs, PK and EC, to discuss consumers’ insights and predict PI for artificial meat products in Taiwan. The following are the detailed explanations of these variables. The research framework is represented in Figure 1.

#### 2.1.1. Theory of Planned Behavior (TPB)

According to the TPB, ATT, SNs, and PBC constitute PI [43] through which behaviors can be directly predicted [44]. PI is defined as the “possibility that a consumer will buy a product” and a higher PI indicates a greater probability of purchasing a product [42,45,46,47,48,49]. PI is often used to predict the occurrence of actual behavior. Marketers have long believed that PI is the most accurate predictor of consumer buying behavior [49,50,51]. Ajzen believed that ATT is an evaluation criterion composed of the sum of an individual’s behavioral beliefs and outcome evaluations [44,52]. Studies have revealed that ATT toward organic foods [53,54] and green products [21,46,55,56,57] have an impact on consumers’ PI. Therefore, this study postulates that consumers’ attitudes toward artificial meat products will have an impact on their PI; thus, H1a is as follows: ATT has a significant positive impact on the intention to purchase artificial meat products.

Ajzen proposed that when individuals are performing a certain behavior, in addition to their own reasoning, they are influenced by family members, friends, colleagues, and the media [44]; this is regarded as SN [46,58,59,60,61]. Scalco et al. showed that SNs significantly influence the PI for organic foods [53]. Therefore, this study speculates that consumers would be influenced by their relatives and friends in deciding whether to buy artificial meat. Thus, the study puts forward H1b: Consumer’s SNs have a significant positive impact on their PI for artificial meat foods. According to Ajzen, PBC refers to how an individual’s perception of their consumption behaviors affects their judgment on the risks and benefits in consuming a product; that is, individuals may be hindered by their past experiences and expectations [43]. It includes the understanding of self-ability, urgent-need perception, and convenience perception [46,61]. Chang et al. found that the PBC significantly affected consumers’ intention to purchase insect foods [20]. Therefore, this study puts forward H1c: Consumer’s PBC has a significant positive effect on the PI for artificial meat products.

#### 2.1.2. Product Knowledge (PK)

PK is defined as consumers’ awareness of specific information about a particular product [62]. Previous studies have shown that prior knowledge affects the processing of information, and PK determines whether consumers choose to buy a product [26]. Different levels of PK may cause consumers to make different decisions. Consumers with a higher level of PK are more familiar with the product, so they are more likely to use intrinsic cues to evaluate product attributes and quality; those with a lower level of PK are more likely to use extrinsic cues to make their choices. The study results reveal that PK has a significant positive impact on PI [26,27,28,63]. Therefore, this study puts forward H2: Consumers’ PK has a significant positive impact on the PI for artificial meat products.

#### 2.1.3. Environmental Concern (EC)

The NEP Scale measures attitudes of individuals toward the environment [30,31,64,65]. Newton et al. posit that EC does not directly affect PI, but it helps consumers to understand the environmental consequences of purchasing a product [66]. Arısal and Atalar argue that EC affects individuals’ PI [67]. Lee also affirmed that EC is an indicator that can be used to predict consumers’ purchasing behavior with regard to green products [68]. In summary, EC plays a role in PI. Therefore, this study puts forward H3: Consumers’ EC has a significant positive impact on the PI for artificial meat products.

## 3. Materials and Methods

### 3.1. The Design of the Survey

The questionnaire was divided into eight parts. Parts 1–3 comprised of the TPB scale. Within the TPB scale, the ATT scale had five items based on the works of Han et al. and Primmer and Karppinen [46,55]; the SN scale had five items based on the works of Bernath and Roschewitz, Spash et al., and Han et al. [46,58,61]; and PBC had five items based on the works of Bernath and Roschewitz, Spash et al., and Han et al. [46,58,61]. Part 4 included the PK scale, which had six items based on the works of Pieniak et al., and Piha et al. [28,63]. Part 5 measured the EC of consumers; it was based on the study by Dunlap and Paul et al. and had 13 items in total [21,31]. Part 6 comprised of the PI scale, which had five items in total, based on the works of Chen, Han et al., and Liang and Lim [45,46,48]. Cronbach’s α for all the scales of the present study was greater than 0.70, indicating that the measurement tools of this study were reasonable. The first six parts mentioned above were all measured using a 7-point Likert scale on attitude, ranging from strongly disagree (1) to strongly agree (7). Part 7 measured the WTP of the respondents. The scale was based on the work of Kang et al. [41], in which consumers were asked what additional fee they would like to pay for a vegetarian meal when they go to a fast food restaurant. The options included 0%, 1%–5%, 6%–10%, and 11% or more. Part 8 was designed to collect the basic data of respondents (i.e., gender, age, religious beliefs, and food culture).

### 3.2. Data Collection

In determining sample size, Yamane [69] provides a simplified formula to calculate sample sizes, assume maximum variability (*p* = 0.5) and desire a 95% confidence level, the minimum sample of the study is 400. Moreover, Comrey and Lee [70] also provides the guidelines to assess the adequacy of the total sample size. They noted that samples of size 100 can give more than adequate reliability correlation coefficients. Respondents were selected based on purposive sampling, with the restriction that they were the main person for purchasing vegetarian burgers. We made a preliminary statement at the beginning of the questionnaire “The questionnaire does not involve any commercial interest and is only for academic research. The results of the questionnaire are also confidential. Thank you for your participation”. In this study, a total of 500 interviews were completed with residents aged 18 years older, and 486 questionnaires were recovered. After eliminating unanswered and incomplete questionnaires, a total of 436 valid questionnaires were procured, with an effective response rate of 87.2%, from 234 females (53.7%) and 202 males (46.3%). With regard to age, 83 respondents were under the age of 20 years (19.0%), 80 respondents were 21–30 years old (18.3%), and 76 respondents were 51–60 years old (17.4%). In terms of religious belief, there were 309 respondents who follow Buddhism and Taoism (70.9%). With regard to food culture statistics, 374 respondents were nonvegetarians (85.8%) and 62 respondents were vegetarians (14.2%). With respect to WTP, 249 respondents (57.1%) were willing to pay an additional 1%–5%, 159 respondents (36.5%) were not willing to pay any additional sum, and 28 respondents (6.4%) were willing to pay an additional 6%–10% (6.4 %).

### 3.3. Statistical Analysis

The theoretical framework was analyzed by employing Statistical Package for Social Science and Analysis of Moment Structure (AMOS) Version 20 (IBM Corp, New York, NY, USA). Two structural equation modeling (SEM) study models were used [71]. A measurement model and a structural model were employed to test the validity and reliability, respectively; the former and the latter were used for testing the model fit and for hypothesis testing, respectively.

The second stage of the data analysis involved using a single-bound dichotomous choice (SBDC) model to analyze the determinants for consumers’ WTP for vegetarian burgers. The collected data were encoded before entered into LIMDEP8.0/NLOGIT4.0 econometric software for analysis. The model for WTP is *Y*_i_^*^ = *X*_i_
*β* + *ε*_i_, where *Y*_i_^*^ is willingness to pay variable, *ε* is a zero-mean error term, and the plan and individual characteristics are summarized into 1 × K vector *X*_i_. The statistical model for the observables expresses the likelihood that respondent will agree on proposed amount, given the plan and individual characteristics. That probability is *Pr* (*Y*_i_^*^ ≥ *C*_i_|*X*_i_) = 1 − *G* (*G*_i_|*X*_i_), where *G* is distribution function of *Y*^*^. *Y*^*^ is assumed to be normal or logistic. The probability of “yes” to a payment of $C is *Pr* (*Y*_i_^*^ = 1) = *E* (*Y*_i_^*^) = *Pr* (*Y*_i_^*^ ≥ *C*_i_) = 1 − *F* ((*C*_i_ − *X*_i_*β*)/*σ*), where *Y*_i_ takes on a value of one if respondent accepts the offer C_i_ (and zero otherwise), *F* is distribution function of *ε*/*σ*, and *σ* is scale parameter of distribution of *Y*^*^. The probability of declining to pay proposed amount is *Pr* (*Y*_i_^*^ ≤ *C*_i_) = F ((*C*_i_ − *X*_i_
*β*)/*σ*) [72].

## 4. Results and Discussion

### 4.1. Measurement Model: Reliability and Validity

The results of the reliability and validity analysis of each variable are shown in Table A1. According to Nunnally, each variable has a high reliability value if the Cronbach’s α coefficient is greater than 0.7 and has a low reliability value if the Cronbach’s α coefficient is lower than 0.35, and thus should be rejected [73]. As for the construct validity, if the factor loading of each variable is higher than 0.5, it means that the item possesses construct validity [74]. The overall reliability of the questionnaires in this study was greater than 0.7, indicating that the questionnaire data have high reliability. The average variance extracted (AVE) and the construct reliability (CR) also matched the standard values. The mean, standard deviation, and correlations among the constructs are presented in Table 1.

### 4.2. Structural Model: Goodness-of-Fit Statistics

The analysis began with a confirmatory factor analysis (CFA) using AMOS 20.0. The measurement model contained five latent constructs (Figure 2). After an initial CFA analysis, the revised model exhibited an appropriate level of model fit: χ^2^/df = 2.379, root mean square error of approximation (RMSEA) = 0.056, goodness-of-fit index (GFI) = 0.931, normalized fit index (NFI) = 0.937, adjusted goodness-of-fit index (AGFI) = 0.967, comparative fit index (CFI) = 0.903, parsimonious normed fit index (PNFI) = 0.672, and root mean square residual (RMR) = 0.035. All the values for composite reliability, which ranged from 0.792 to 0.916, clearly exceeded the minimum threshold of 0.60. This result supported internal consistency among the items for each construct. In addition, all AVE values in the present study were greater than 0.50, thus supporting convergent validity. Finally, the degree of discriminant validity was acceptable, and the AVE value for each study variable clearly exceeded the squared value for its correlations with other study variables [74].

### 4.3. Hypothesis Testing

The proposed model was evaluated by running SEM with the maximum likelihood estimation method. The findings indicate that our proposed model had a satisfactory predictive ability in outcome variables. The path analysis result and verification of the hypotheses are demonstrated in Figure 2.

The study’s results revealed that consumers’ ATT had a significant positive impact on the PI for vegetarian burgers (β = 0.174, *p* < 0.001); thus, H1a is valid; that is, when consumers think that vegetarian burgers are healthy or they themselves possess environmental and social responsibilities, they have a more positive attitude toward vegetarian burgers and are more likely to buy them. This result is consistent with that of previous studies [25,54,56,57]. Second, the study results revealed that consumers’ SNs had no significant relationship with the PI for vegetarian burgers (β = 0.440, *p* > 0.05), therefore H1b is not tenable; that is, consumers will not buy vegetarian burgers because of the opinions of their family members, friends, and colleagues. This result is consistent with that of the study conducted by Tan et al., which demonstrated no significant relation between the SNs of consumers and their willingness to buy energy-saving appliances [75]. However, Scalco et al. concluded that consumer’s SNs can significantly affect their PI for organic foods [53]. This study infers that even if Taiwanese newspapers or magazines report the benefits of artificial meat foods, they are still novel products, therefore consumers are less likely to use them because of others’ opinions. With regard to home appliances, consumers are familiar with products and have their own favorite brands, thus they are less likely to consult with others; this may explain the differences in study results.

Consumers’ PBC had a significant positive effect on their PI for fast food burgers (β = 0.014, *p* < 0.001), therefore H1c is tenable; that is, when consumers think that the purchase of vegetarian burgers is the right decision, their PI will increase. This result is in accordance with that of previous studies [20,22]. In addition, consumers’ PK for vegetarian burgers had no significant relation with their PI (β = 0.235, *p* > 0.05), therefore H2 is not tenable; this result is inconsistent with that of previous studies [26,27,28,63]. This study infers that artificial meat foods are still relatively new to Taiwanese consumers, and each consumer has different PK, which affects their PI for vegetarian burgers. Finally, consumers’ EC had a significant positive effect on their PI for vegetarian burgers (β = 0.146, *p* < 0.001), thus H3 is tenable; that is, the more positive the consumers’ EC is, the more likely they are to purchase vegetarian burgers. This result is consistent with that of previous studies revealing that consumers’ EC significantly affects their PI [21,32,33,34]. On the basis of these findings, H1a, H1c, and H3 are supported, but H1b and H2 are not.

### 4.4. WTP Analysis

A cross-over analysis of the respondents’ socioeconomic characteristics and their WTP were conducted. The results are shown in Table 2. Except for gender, WTP for fast food burgers varied with differences in age, religion, and food culture. In terms of age, respondents over 51 years of age had significantly higher WTP (100%) than other age groups. Respondents who followed Buddhism and Taoism had significantly higher WTP (76.7%) than those who followed or did not follow other religions (24.7%). Therefore, this study infers that religion, as a factor, has a significant influence. Vegetarians have significantly higher WTP (91.9%) than nonvegetarians (8.1%). Thus, this study infers that this population group may have a dietary preference for health preservation, environmental protection, and ahimsa.

To analyze the factors influencing consumers’ willingness to pay a premium price for vegetarian burgers, SBDC model was employed. Table 3 shows the results of the SBDC model. The coefficients associated with the bid levels are positive, reflecting a positive relationship between the bid amount/price premium and the likelihood of acceptance. We found that female, Buddhism and Taoism, vegetarian respondents were more willing to pay for vegetarian burgers. Age was inversely related to WTP. This study concludes that the average WTP value for vegetarian burgers is TWD $1358 (US $45.14; EUR$41.40) per person per year.

## 5. Conclusions

### 5.1. Concluding Remarks

The results of this study reveal that consumers’ ATT, PBC, and EC have significant impact on their PI for vegetarian burgers. Therefore, this study suggests that food or catering companies should focus on corporate environmental and social responsibilities and raise awareness among their customers that artificial meat products cause less carbon emissions than pork and chicken meat, and therefore are environment-friendly, by conducting lectures and through media promotions to strengthen consumers’ environmental awareness. Further, the findings indicate that consumers’ SNs and PK have no significant relation with their PI for vegetarian burgers. Therefore, this study suggests that food or catering firms should organize cooking-related training activities, such as cooking in a manner familiar to consumers [76], use artificial meat in products familiar to consumers [75], and promote methods such as tasting, to strengthen consumers’ product experience and interest in artificial meat products, thereby increasing their PI. In addition, Taiwan’s current catering business and channels of selling artificial meat foods are still not extremely popular, thus, people are not familiar with artificial meat foods. Therefore, it is recommended that relevant/related industrial personnel should expand their marketing channels for greater convenience of purchasing artificial meat foods.

Furthermore, the study results reveal that vegetarians have significantly higher WTP additional charges for vegetarian burgers than nonvegetarians. Moreover, in recent years, the number of people becoming vegetarians has increased because of awareness regarding environmental conservation and protection, carbon reduction, health, and other factors. Therefore, it is recommended that food or catering firms launch vegetarian tourism and courses to increase business opportunities and achieve the goal of reducing carbon emissions.

### 5.2. Limitations of the Study and Scope for Future Research

Issues related to environmental ecology and sustainable development are globally relevant topics and have increasingly gained attention from consumers. However, artificial meat products, which are environment-friendly and aid in maintaining ecological balance and promoting sustainable development as organic foods and green products, are still not widely accepted by consumers, despite newspapers and magazines highlighting their merits. Therefore, future studies must analyze the perceptions and viewpoints of consumers belonging to school level, living in different regions between urban and non-urban responses, different cultural backgrounds (e.g., Muslims) on artificial meat foods. In addition, the study recommends including other variables such as information asymmetry and health awareness to explore whether they affect the PI to complete the research framework. Furthermore, it has been mentioned in a previous study that peer communication on social media can significantly affect PI [77]. Therefore, in the future, social media can also be used to investigate the factors influencing consumers’ acceptance of artificial meat foods.

Further, there has been much criticisms center largely on the validity and utility of the TPB. Given the complex range of conscious and unconscious factors that influence behaviour, it does not seem feasible for a single theoretical framework comprising a limited set of constructs to sufficiently explain or predict complex human behaviors, which is consistent with the views expressed by Sniehotta et al. [78]. Several studies proposed that value is a more basic social cognition than attitude [79], and through the establishment of value, eating habits and long-term behaviors might be changed permanently [80]. Therefore, there has been a growing number of research studies with the increasing usage of Value–Attitude–Behavior Model (VAB) in consumers’ purchase behavior. Honkanen et al. [81] combined moral cognition with this shift, exploring whether consumers’ moral values affect choice of organic food due to environmental factors and animal welfare. Kang, Jun, and Arendt [82] applied the VAB Model to investigate purchase intention on low-calorie food. Jun et al. [80] also studied the effects of health value on healthful food selection intention with VAB. Our results suggest that broader and more integrative approaches to explaining dietary behaviour may be more successful and consequently more useful in the development of behavioral change strategies.

## Figures and Tables

**Figure 1 foods-09-00745-f001:**
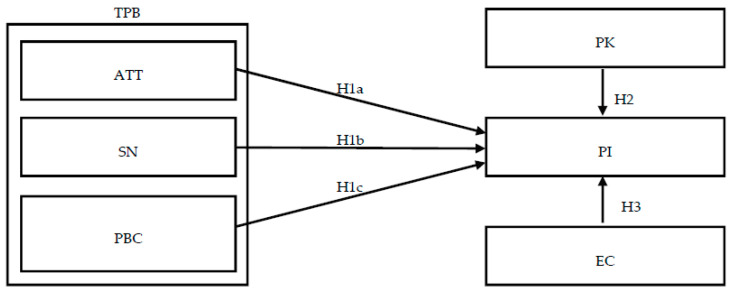
Conceptual model. TPB, theory of planned behavior; ATT, attitude; SN, subjective norm; PBC, perceived behavioral control; PK, product knowledge; EC, environmental concern; PI, purchase intention; H, Hypothesis.

**Figure 2 foods-09-00745-f002:**
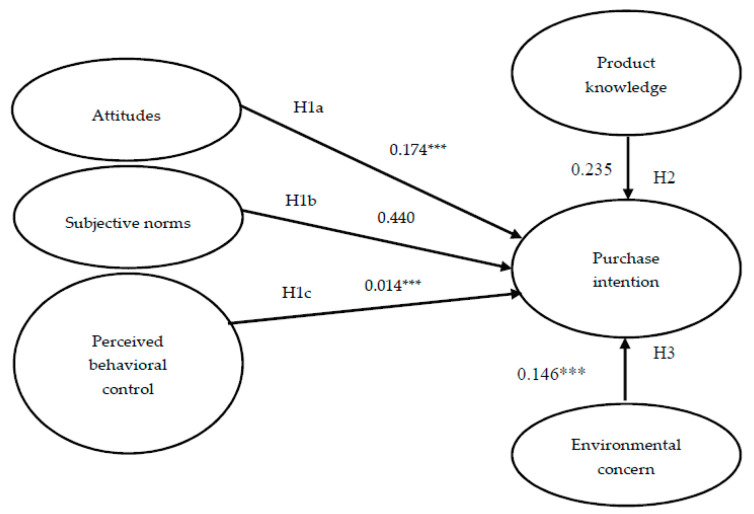
Paths within the hypothesis model. Note: *** *p* < 0.001; solid lines denote established hypotheses.

**Table 1 foods-09-00745-t001:** Mean, standard deviation (SD), and correlations of constructs.

Construct	Mean	SD	1	2	3	4	5	6
1. ATT	4.74	1.57	1.00					
2. SN	4.58	1.28	0.67	1.00				
3. PBC	6.32	0.74	0.56	0.36	1.00			
4. PK	4.39	1.40	0.48	0.30	0.33	1.00		
5. EC	5.28	1.74	0.42	0.28	0.25	0.42	1.00	
6. PI	4.61	1.40	0.37	0.32	0.36	0.39	0.47	1.00

Notes: ATT, attitude; SN, subjective norm; PBC, perceived behavioral control; PK, product knowledge; EC, environmental concern.

**Table 2 foods-09-00745-t002:** Cross-over of analysis of demographic variables and willingness to pay (WTP).

WTP		0%	1–5%	6–10%	Total	χ^2^
Gender	Male	84	107	11	202	4.390
	Female	75	142	17	234	
	Subtotal	159	249	28	436	
Age	Under 20 years	39	41	3	83	137.779 ***
	21–30 years old	32	45	3	80	
	31–40 years old	46	20	3	69	
	41–50 years old	42	19	9	70	
	51–60 years old	0	71	5	76	
	61 years and above	0	53	5	58	
	Subtotal	159	249	28	436	
Religious beliefs	Buddhism and Taoism	72	211	26	309	88.281 ***
	Nonreligious	14	15	1	30	
	Others	73	23	1	97	
	Subtotal	159	249	28	436	
Food culture	Nonvegetarian	154	203	17	374	34.105 ***
	Vegetarian	5	46	11	62	
	Subtotal	159	249	28	436	

Note: *** *p* < 0.001.

**Table 3 foods-09-00745-t003:** Single-bound dichotomous choice (SBDC) results.

Variable	Coefficient	Standard Error	*z* Value
Constant	28.768324 ***	7.466539	5.476
Female	0.468152 ***	0.386692	4.575
Age	−0.376432 ***	0.016786	−8.752
Buddhism and Taoism	0.478654 **	0.298675	2.637
Vegetarian	0.498655 **	0.185433	3.465
log(bid)	0.686456	0.356517	−5.431
Log-likelihood		−265.74	
Mean WTP	1.358		

Note: *** *p* < 0.001; ** *p* < 0.05; TWD $1,358 (=US $45.14 = EUR$41.40).

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
