# Peer review of "Exploring Consumers’ Purchase Intention of an Innovation of the Agri-Food Industry: A Case of Artificial Meat"

_foods, 2020, doi:10.3390/foods9060745_

Round 1

Reviewer 1 Report

This study used a wide extended approach to examine the purchase intention of a more sustainable protein alternative of meat in Taiwan, vegetarian meat. Authors used a combination of behavioral theories to determine the factors that can predict a higher purchase intention of a vegetarian meat such as the Theory of planned Behavior (TPB), the new environmental paradigm and the product knowledge. Despite this well–known theoretical background, authors explore an interesting proposal of the food system faced to reduce the green gas emission.  However, the manuscript can be improved in several aspects.

In general, the manuscript is understandable, but I suggest to authors improve the writing style in order to improve the reading flow. 

The title is not at all appropriate. The main issue of this manuscript is the purchase intention of an innovation of the agri-food industry. Authors can use it to develop a new title.

Abstract

Authors should be following the journal guidance about the abstract. This is The abstract should be a single paragraph and should follow the style of structured abstracts, but without headings”. In this line, I suggest drop the headings.

Moreover, there are a misconceptions relate to the definition of purchase intention measure thought the theory of planned behavior and willingness to pay.  For example, authors said “Based on the theory of planned behavior, the willingness … to pay for fat food burgers is discussed”. But in fact, authors used the theory of planned behavior to explain the purchase intention not the willingness to pay for vegetarian burgers. And the willingness to pay was explained thought the consumers characteristics.

Please do not interchange the meaning of these two concepts. In consumer behavior, there is wide extended evidence that highlighted that high purchase intention do not always means high purchase behavior (usually measured thought willingness to pay or buy) and vice versa. In fact, in the literature this gap is not stilling fill yet.

Other element that author should change is the conclusions. I suggest move this sentences to results.

Introduction

Introduction included most of the key points that follows the objective. However, I suggest improve the writing style in order to increase the effectiveness, clearness and organization of these key points.

Other issues in this section

Paragraph from line 68 to 70 can be dropped from this section because the detail of the theoretical approach is in the section 2.

In line 71, please change the expression “research area of consumers’ willingness to buy” to research area of consumer’s purchase intention.

From line 66 to 67 a reference is missing at the end of the sentence.

In line 78, I am disagreeing when the authors said that the study fill a gap in the literature. When we talk of fill a gap in the literature, for example, we expect to find a new approach that explained in major measure the consumer behavior, but the manuscript does not show this results. I think that the main contribution of the paper is providing the first findings regarding the consumer behavior of a polemic but sustainable protein alternative of meat. 

Authors said that the price is a decisive factor of purchase intention, but the literature presented is related to the purchase behavior. Again this is a misconception.

I suggest drop the sentence from line 98 to 99 and sentence from line 103 to 104

Sentence from line 110 to 112 can be moved to methods section.

Review of literature and hypothesis development

In this section, the paragraphs are short. In consequence, I think the subheadings are unnecessary.

Please, check reference style in line 130.

Materials and Methods

Subheading 3.1 include figure 1 can be move to review of literature section.

There is missing information about the sample. For example, authors do not explain how the sample was recruited, ethical protocols, what was the strategy follows to stratify the sample? There were restrictions of age to participate in the survey, legal age?

I suggest include in a table the main sociodemographic characteristic of the sample. Also would be interesting explained if the sample is representative of the Taiwan population.

Authors do not explain with details the technique used in the CVM, the econometrical model and the statistical package used to analyze the data.

Results and Discussion

I suggest move table to an appendix section.

Result section explained with detail the goodness of fit of the TPB. Authors only present a consumer profile according the WTP, but omitted the econometrical model of the CVM an also the regression analysis. In this line, I suggest extend the result section with the inclusion of the regression analysis of the CVM.

Moreover, the results of the purchase intention were developed in the discussion section. I suggest to the authors merge both section or move the results to their correspondent section and kept the discussion section.

I suggest drop table 3 and move part of the information to complete figure 2.

Conclusion

Despite the evidence providing for the findings of this study, authors do not considered the limitations of the theoretical framework used (i.e. TPB). What other consumer behavior theories can be used to understand the purchase intention and purchase behavior of this type of product?

Author Response

Thanks very much for your valuable review comments and suggestions. We believe your comments are appropriate and useful for us to greatly improve the manuscript quality. We now has revised this manuscript according to your review comments and instructions. Point-by-point response is noted in the attached document.

Comments and Suggestions for Authors:
This study used a wide extended approach to examine the purchase intention of a more sustainable protein alternative of meat in Taiwan, vegetarian meat. Authors used a combination of behavioral theories to determine the factors that can predict a higher purchase intention of a vegetarian meat such as the Theory of planned Behavior (TPB), the new environmental paradigm and the product knowledge. Despite this well–known theoretical background, authors explore an interesting proposal of the food system faced to reduce the green gas emission. However, the manuscript can be improved in several aspects.
In general, the manuscript is understandable, but I suggest to authors improve the writing style in order to improve the reading flow.

1.The title is not at all appropriate. The main issue of this manuscript is the purchase intention of an innovation of the agri-food industry. Authors can use it to develop a new title.

Revision Feedbacks: Thanks for your kind suggestions, we have revised it as follows:
Exploring consumers’ purchase intention of an innovation of the agri-food industry: A case of artificial meat

2.Authors should be following the journal guidance about the abstract. This is “The abstract should be a single paragraph and should follow the style of structured abstracts, but without headings”. In this line, I suggest drop the headings.

5.Other element that author should change is the conclusions. I suggest move this sentences to results.

Revision Feedbacks: Thanks for your kind suggestions, we have revised it (please see page 1 in abstract for detail)

3.Moreover, there are a misconceptions relate to the definition of purchase intention measure thought the theory of planned behavior and willingness to pay. For example, authors said “Based on the theory of planned behavior, the willingness … to pay for fat food burgers is discussed”. But in fact, authors used the theory of planned behavior to explain the purchase intention not the willingness to pay for vegetarian burgers. And the willingness to pay was explained thought the consumers characteristics.

4.Please do not interchange the meaning of these two concepts. In consumer behavior, there is wide extended evidence that highlighted that high purchase intention do not always means high purchase behavior (usually measured thought willingness to pay or buy) and vice versa. In fact, in the literature this gap is not stilling fill yet.

Revision Feedbacks: Thank you for your in-depth analysis and practical advice, we have revised it (please see page 1 in abstract for detail)

6.Introduction included most of the key points that follows the objective. However, I suggest improve the writing style in order to increase the effectiveness, clearness and organization of these key points.

Revision Feedbacks: Thank you for your advice. We have made more clearly description of the Introduction. (please see page 3 in Introduction for detail)

7.Paragraph from line 68 to 70 can be dropped from this section because the detail of the theoretical approach is in the section 2.

Revision Feedbacks:Thanks for your suggestions, we have deleted it.

8.In line 71, please change the expression “research area of consumers’ willingness to buy” to research area of consumer’s purchase intention.

Revision Feedbacks: Thank you for your in-depth analysis and practical advice, we have revised “research area of consumers’ willingness to buy” to “research area of consumer’s purchase intention”.

9.From line 66 to 67 a reference is missing at the end of the sentence.

Revision Feedbacks: Thank you for your in-depth analysis and practical advice, we have revised it.

10.In line 78, I am disagreeing when the authors said that the study fill a gap in the literature. When we talk of fill a gap in the literature, for example, we expect to find a new approach that explained in major measure the consumer behavior, but the manuscript does not show this results. I think that the main contribution of the paper is providing the first findings regarding the consumer behavior of a polemic but sustainable protein alternative of meat.

Revision Feedbacks: Thank you for your helpful advice. We have already deleted “…the study fill a gap in the literature….” and added “…..the main contribution of the paper is providing the first findings regarding the consumer behavior of a polemic but sustainable protein alternative of meat”.
Thank you for your in-depth analysis and practical advice, we have revised it.

11.Authors said that the price is a decisive factor of purchase intention, but the literature presented is related to the purchase behavior. Again this is a misconception.

Revision Feedbacks: We apologize for the error. Thank you for your in-depth analysis and practical advice, we have revised it

12.I suggest drop the sentence from line 98 to 99 and sentence from line 103 to 104

Revision Feedbacks: Thanks for your suggestions, we have dropped the sentence from line 98 to 99 and sentence from line 103 to 104.

13.Sentence from line 110 to 112 can be moved to methods section.

Revision Feedbacks: Thank you for your helpful advice. To address your concerns, we have made changes by moving the part in the section of “line 110 to 112 ” to the section of “methods ”.

14.In this section, the paragraphs are short. In consequence, I think the subheadings are unnecessary.

Revision Feedbacks: Thanks for your suggestions, we have deleted the subheadings.

15.Please, check reference style in line 130.

Revision Feedbacks: Thanks for your suggestions, we have revised it.

16.Subheading 3.1 include figure 1 can be move to review of literature section.

Revision Feedbacks: Thank you for your helpful advice. To address your concerns, we have made changes by moving the part in the section of “Subheading 3.1 include figure 1 ” to the section of “literature ”.

17.There is missing information about the sample. For example, authors do not explain how the sample was recruited, ethical protocols, what was the strategy follows to stratify the sample? There were restrictions of age to participate in the survey, legal age?

Revision Feedbacks: Thank you for your in-depth analysis and practical advice. This section has been rewritten as;
3.2 Data Collection
In determining sample size, Yamane [69] provides a simplified formula to calculate sample sizes, assume maximum variability (p = 0.5) and desire a 95% confidence level, the minimum sample of the study is 400. Moreover, Comrey & Lee [70] also provides the guidelines to assess the adequacy of the total sample size. They noted that samples of size 100 can give more than adequate reliability correlation coefficients. Respondents were selected based on purposive sampling, with the restriction that they were the main person for purchasing vegetarian burgers. We made a preliminary statement at the beginning of the questionnaire “The questionnaire does not involve any commercial interest and is only for academic research. The results of the questionnaire are also confidential. Thank you for your participation.” In this study, a total of 500 interviews were completed with residents aged 18 years older……..

18.I suggest include in a table the main sociodemographic characteristic of the sample. Also would be interesting explained if the sample is representative of the Taiwan population.

Revision Feedbacks:Thank you for your in-depth analysis and practical advice. This section has been rewritten as; 486 questionnaires were recovered. After eliminating unanswered and incomplete questionnaires, a total of 436 valid questionnaires were procured, with an effective response rate of 87.2%, from 234 females (53.7%) and 202 males (46.3%). As regards age, 83 respondents were under the age of 20 years (19.0%); 80 respondents were 21–30 years old (18.3%); and 76 respondents were 51–60 years old (17.4%). In terms of religious belief, there were 309 respondents who follow Buddhism and Taoism (70.9%). With regard to food culture statistics, 374 respondents were nonvegetarians (85.8%) and 62 respondents were vegetarians (14.2%). As regards WTP, 249 respondents (57.1%) were willing to pay an additional 1%–5%, 159 respondents (36.5%) were not willing to pay any additional sum, and 28 respondents (6.4%) were willing to pay an additional 6%–10% (6.4 %).

19.Authors do not explain with details the technique used in the CVM, the econometrical model and the statistical package used to analyze the data.

Revision Feedbacks: Thanks for your suggestions. This section has been rewritten as; The second stage of the data analysis involves by using a single-bound dichotomous choice model to analyze the determinants for consumers’ WTP for vegetarian burgers. The collected data were encoded before entered into LIMDEP8.0/NLOGIT4.0 econometric software for analysis.

20.I suggest move table to an appendix section.

Revision Feedbacks: Thank you for your helpful advice. To address your concerns, we have made changes by moving the part in the section of “table 1 ” to the section of “appendix ”.

21.Result section explained with detail the goodness of fit of the TPB. Authors only present a consumer profile according the WTP, but omitted the econometrical model of the CVM an also the regression analysis. In this line, I suggest extend the result section with the inclusion of the regression analysis of the CVM.

Revision Feedbacks: Thanks for your suggestions. This section has been rewritten as; CVM data can be modeled as follows [75]: The model for WTP is ?? = ?? ? + ?? Where, ?? is willingness to pay variable, ? is a zero-mean error term, and the plan and individual characteristics are summarized into 1×K vector ??. The statistical model for the observables expresses the likelihood that respondent will agree on proposed amount, given the plan and individual characteristics. That probability is ??(?? ≥ ?? |?? ) = 1 − ?(?? |?? ), where G is distribution function (cdf) of ? . ? is assumed to be normal or logistic. The probability of “yes” to a payment of $C is Pr(?? = 1) = ?(?? ) = ??(?? ≥ ?? ) = 1 − ?((?? − ???)/?), where ?? takes on a value of one if respondent accepts the offer Ci (and zero otherwise), F is the cdf of ?/? , and ? is scale parameter of distribution of ? . The probability of declining to pay proposed amount is Pr(?? ≤ ?? ) = F((?? − ?? ?)/?) For estimation, LIMDEP8.0 / NLOGIT4.0 econometrics software was employed and results from the basic and extended multinomial logit models as well as the random parameter logit model. This study concludes that the average WTP value for vegetarian burgers is NT$1358 per person per year.

22.Moreover, the results of the purchase intention were developed in the discussion section. I suggest to the authors merge both section or move the results to their correspondent section and kept the discussion section.

Revision Feedbacks: Thanks for your suggestions, we have merge both results and discussion section

23.I suggest drop table 3 and move part of the information to complete figure 2.

Revision Feedbacks: Revision Feedbacks Thanks for your suggestions, we have dropped table 3.

24. Despite the evidence providing for the findings of this study, authors do not considered the limitations of the theoretical framework used (i.e. TPB). What other consumer behavior theories can be used to understand the purchase intention and purchase behavior of this type of product?

Revision Feedbacks: We appreciate your keen observation. First, we added detailed descriptions of the results with their implications which were not included in the discussion section in the first place. Also, additional implication was added to provide more insight from the methods, results and discussion section. In addition, a new citation was added to compare with a result of a previous study. Finally, amendment of study result was employed to offer a sufficient detail of the discussion content correctly. This section has been rewritten as; Further, there has been much criticisms centre largely on the validity and utility of the TPB. Given the complex range of conscious and unconscious factors that influence behaviour, it does not seem feasible for a single theoretical framework comprising a limited set of constructs to sufficiently explain or predict complex human behaviours. Consistent with the views expressed by Sniehotta et al. [78]. Several studies proposed that value is more basic social cognitions than attitude [79], and through the establishment of value, eating habit and long-term behavior might be changed for long [80]. Therefore, there has been a growing number of research studies with the increasing usage of Value-Attitude-Behavior Model (VAB) in consumers’ purchase behavior. Honkanen et al. [81] combined moral cognition with this shift, exploring whether consumers' moral values affect choice of organic food due to environmental factors, animal welfare. Kang, Jun and Arendt [82] applied VAB Model to investigate purchase intention on low-calorie food. Jun et al. [80] also studied the effects of health value on healthful food selection intention with VAB. Our results suggest that broader and more integrative approaches to explaining dietary behaviour may be more successful and consequently more useful in the development of behavioural change strategies.

Reviewer 2 Report

The main question addressed by the research is to assess the acceptability of artificial meat to the consumer. The topic itself is interesting and the text is clear and easy to read. The type of analysis is a survey aimed at investigating consumer reactions. The topic is original, the authors have developed it in a mediocre way. However, the work adds little to the subject area compared with other published material, because it does not identify a variable other than the existing one.

The title don't report the topic of the paper : consumer behavior about purchase of artificial meat.

The conclusions are in line with what has been described but surely the experimental design is limited and therefore the conclusions are also limited (for example an analysis on the school sector is missing).

pag. 192 It's not available the criteria about the distribution of 500 questionnaires

pag. 195 the authors have to explain why the haven't expected the variable " school level" , but only gender age, religious belief and food culture.

Author Response

Thanks very much for your valuable review comments and suggestions. We believe your comments are appropriate and useful for us to greatly improve the manuscript quality. We now has revised this manuscript according to your review comments and instructions. Point-by-point response is noted in the attached document.

 Comments and Suggestions for Authors
The main question addressed by the research is to assess the acceptability of artificial meat to the consumer. The topic itself is interesting and the text is clear and easy to read. The type of analysis is a survey aimed at investigating consumer reactions. The topic is original, the authors have developed it in a mediocre way. However, the work adds little to the subject area compared with other published material, because it does not identify a variable other than the existing one.

1.The title don't report the topic of the paper : consumer behavior about purchase of artificial meat.

Revision Feedbacks: Thanks for your kind suggestions and we have revised it as follows:
Exploring consumers’ purchase intention of an innovation of the agri-food industry: A case of artificial meat

2.The conclusions are in line with what has been described but surely the experimental design is limited and therefore the conclusions are also limited (for example an analysis on the school sector is missing).

Revision Feedbacks: We appreciate your keen observation. First, we added detailed descriptions of the results with their implications which were not included in the discussion section in the first place. Also, additional implication was added to provide more insight from the methods, results and discussion section. In addition, a new citation was added to compare with a result of a previous study. Finally, amendment of study result was employed to offer a sufficient detail of the discussion content correctly. This section has been rewritten as; Further, there has been much criticisms centre largely on the validity and utility of the TPB. Given the complex range of conscious and unconscious factors that influence behaviour, it does not seem feasible for a single theoretical framework comprising a limited set of constructs to sufficiently explain or predict complex human behaviours. Consistent with the views expressed by Sniehotta et al. [78]. Several studies proposed that value is more basic social cognitions than attitude [79], and through the establishment of value, eating habit and long-term behavior might be changed for long [80]. Therefore, there has been a growing number of research studies with the increasing usage of Value-Attitude-Behavior Model (VAB) in consumers’ purchase behavior. Honkanen et al. [81] combined moral cognition with this shift, exploring whether consumers' moral values affect choice of organic food due to environmental factors, animal welfare. Kang, Jun and Arendt [82] applied VAB Model to investigate purchase intention on low-calorie food. Jun et al. [80] also studied the effects of health value on healthful food selection intention with VAB. Our results suggest that broader and more integrative approaches to explaining dietary behaviour may be more successful and consequently more useful in the development of behavioural change strategies.

3.pag. 192 It's not available the criteria about the distribution of 500 questionnaires.

Revision Feedbacks: Thank you for your in-depth analysis and practical advice. This section has been rewritten as;
3.2 Data Collection
In determining sample size, Yamane [69] provides a simplified formula to calculate sample sizes,
assume maximum variability (p = 0.5) and desire a 95% confidence level, the minimum sample of the
study is 400. Moreover, Comrey & Lee [70] also provides the guidelines to assess the adequacy of the
total sample size. They noted that samples of size 100 can give more than adequate reliability
correlation coefficients. Respondents were selected based on purposive sampling, with the restriction
that they were the main person for purchasing vegetarian burgers. We made a preliminary statement
at the beginning of the questionnaire “The questionnaire does not involve any commercial interest and is
only for academic research. The results of the questionnaire are also confidential. Thank you for your participation.” In this study, a total of 500 interviews were completed with residents aged 18 years
older……..

4.pag. 195 the authors have to explain why the haven't expected the variable " school level" , but only gender age, religious belief and food culture.

Revision Feedbacks: Thanks for your suggestions. One of the study’s main limitations is the use of a purposive sampling technique, which does not allow the findings to be generalized to the overall population of the customers’ attitude toward artificial meat foods in Taiwan. Future research could also investigate the effect of other customer-related constructs, such as school level, living in different regions between urban and non-urban responses.

Round 2

Reviewer 1 Report

In general, authors answered my comments. However, there are two points that can be improves.

  • When I suggest move figure 1 to the theoretical framework, my intention was that this figure works such as summary of this subheading. In this line, authors can be reorganizing this subheading as follows:

2.1 Theoretical framework

2.1.1 TPB

2.1.2 PK

2.1.3 EC

Figure 1

  • The CVM specification (presented in page 8, lines 298-308) is more appropriate of methods section. Instead, authors can be explain the output model estimation (B coefficient, standard error, z-value, p-value) and then indicate the mean WTP. On another hand, authors should include the standard error and the p-value of the WTP. Finally, besides presenting WTP in NT$, authors should presents the WTP in euro or dollars.

Author Response

Thanks very much for your valuable review comments and suggestions. We believe your comments are appropriate and useful for us to greatly improve the manuscript quality. We now has revised this manuscript according to your review comments and instructions. Point-by-point response is noted in the attached document.

Comments and Suggestions for Authors

In general, authors answered my comments. However, there are two points that can be improves.

1.When I suggest move figure 1 to the theoretical framework, my intention was that this figure works such as summary of this subheading. In this line, authors can be reorganizing this subheading as follows:

2.1 Theoretical framework

2.1.1 TPB

2.1.2 PK

2.1.3 EC

Figure 1

Revision Feedbacks: Thank you for your in-depth analysis and practical advice, we have revised it (please see page 3-4 in Review of Literature and Hypothesis Development for detail)

2.The CVM specification (presented in page 8, lines 298-308) is more appropriate of methods section.

Revision Feedbacks: Thank you for your helpful advice. To address your concerns, we have made changes by moving the part in the section of “lines 298-308 ” to the section of “methods ”.

2.Instead, authors can be explain the output model estimation (B coefficient, standard error, z-value, p-value) and then indicate the mean WTP. On another hand, authors should include the standard error and the p-value of the WTP. Finally, besides presenting WTP in NT$, authors should presents the WTP in euro or dollars.

Revision Feedbacks: Thank you for your in-depth analysis and practical advice. This section has been rewritten as;
To analyze the factors influencing consumers’ willingness to pay a premium price for vegetarian
burgers, SBDC model was employed. Table 3 shows result of SBDC model. the coefficients associated with the bid levels are positively, reflecting a positive relationship between the bid amount / price premium and the likelihood of acceptance. We found that female, Buddhism and Taoism, Vegetarian respondents are more WTP for vegetarian burgers. Age was inversely related to WTP. This study concludes that the average WTP value for vegetarian burgers is TWD $1,358 (=US $45.14=EUR$41.40) per person per year.

Table 3. Single-bound dichotomous choice (SBDC) results.

Variable

Coefficient

Standard Error

z value

Constant

28.768324 ***

7.466539

5.476

Female

0.468152 ***

0.386692

4.575

Age

−0.376432 ***

0.016786

−8.752

Buddhism and Taoism

0.478654 **

0.298675

2.637

Vegetarian

0.498655 **

0.185433

3.465

log(bid)

0.686456

0.356517

−5.431

Log-likelihood

−265.74

Mean WTP

1.358

Note: ***p < 0.001; **p < 0.05; TWD $1,358 (=US $45.14 = EUR$41.40).